# Assessment of Saudi Mother’s Knowledge and Attitudes towards Childhood Diarrhea and Its Management

**DOI:** 10.3390/ijerph18083982

**Published:** 2021-04-09

**Authors:** Sultan Alghadeer, Wajid Syed, Abdulaziz Alhossan, Ziyad Alrabiah, Salmeen D. Babelghaith, Mohamed N. Al Arifi, Abdulrahman Alwhaibi

**Affiliations:** Department of Clinical Pharmacy, College of Pharmacy, King Saud University, Riyadh 11451, Saudi Arabia; salghadeer@ksu.edu.sa (S.A.); alhossan@ksu.edu.sa (A.A.); Zalrabiah@ksu.edu.sa (Z.A.); sbabelghaith@ksu.edu.sa (S.D.B.); malarifi@ksu.edu.sa (M.N.A.A.); aalwhaibi@KSU.EDU.SA (A.A.)

**Keywords:** diarrhea, Saudi Arabia, knowledge, attitude, children, mother

## Abstract

This study evaluates the knowledge, attitude of Saudi mothers towards diarrhea in their children and its management at home. Online cross-sectional validated online surveys, targeting Saudi mothers who are living in Saudi Arabia, are used to collect data from the beginning of March to the end of April 2019. Our results show that a total of 1140 mothers (52.1% of them were housewives) participated in the study. Approximately 40.3% of participating mothers believed that childhood diarrhea is a major problem in the Saudi community; however, almost 23% of the participants were unable to identify any critical sign of severe diarrhea, and around 66% falsely stated that diarrhea is caused by teething. Although 62% of our participating mothers knew about oral rehydration therapy (ORS), only 23.5% of them used it for their children. Adequate knowledge about the critical signs, causes, transmission, prevention, and management of childhood diarrhea should be applied in simple language to communicate the health-related information clearly.

## 1. Introduction

Diarrhea is a common illness that is considered a major threat to children, and it may lead to death in developing countries, particularly amongst children aged up to five years [1,2,3]. Twelve million children are estimated to die in developing countries before the age of five years. Around 70% of those children die because of five medical issues; one of them is diarrhea [1,4]. Despite that age, climate changes, and the use of rotavirus vaccine can contribute to variations in pathogen-causative diarrhea; rotavirus was the most common causative pathogen, especially in unvaccinated children against rotavirus [5]. In Saudi Arabia, rotavirus was noticed in 41.3% to 65.5% of cases causing the child’s diarrhea [6,7]. In addition to microbial-induced diarrhea, diarrhea can result from intolerance of certain kinds of food, particularly lactose-containing milk [8].

Regardless of diarrhea-induced causes, dirty weaning food, improper nourishing practice, absence of clean water, poor hand washing, constrained sterile transfer of waste, poor lodging conditions, and absence of access to satisfactory and moderate social insurance are aggravating factors of diarrhea in children under five years old [9]. A study in Saudi Arabia reported that environmental risk factors associated with diarrhea in children included sewage leakage near the home, eating out after school hours, and utilizing reusable cloths or sponges to dry dishes [10].

Enhancing mothers’ proper knowledge and demonstrating appropriate practice is a key to prevent or halt the progression of diarrhea. However, mothers’ harmful practices, such as nourishment limitation, breastfeeding avoidance, and utilization of inappropriate conventional therapy or wrong prescription, have been reported [11]. In addition, the mothers’ knowledge on the signs of dehydration secondary to diarrhea is poor [2,12]. In most cases, diarrhea can be treated at home by oral rehydration therapy (ORT) that has significantly decreased the mortality related to diarrhea disease [13]. Although this method of treatment is cheap, adequate, reasonable, and safe, few mothers listed that the aim of using (ORS) during diarrhea is to treat dehydration of diarrhea [2,13]. To demonstrate the best home care of children with diarrhea in our community by investigating certain knowledge, attitude, and particular practice toward diarrhea in children, and its management is essential. Therefore, this study evaluates the knowledge and attitudes of mothers towards diarrhea in children and its management at home in Saudi Arabia.

## 2. Methods and Materials

### 2.1. Design and Settings

Online cross-sectional survey-based study targeting mothers who are living in Saudi Arabia were carried out to measure knowledge, attitude, and practice of mothers towards diarrhea and its management in their children at home. The data collection was carried out from the beginning of March to the end of April 2019. Saudi women bearing at least one child below the age of five years were included in the study.

### 2.2. Design of the Questionnaires

The questionnaires for this study were prepared after an extensive literature review from similar studies published in this regard [14,15]. The questionnaires for this study are grouped into four parts. The first part was to collect demographic data (such as the mother’s age, the mother’s education level, the mother’s job, the child’s age, etc.). The second part is the knowledge domain questions with the multiple-choice options discussing subjects like signs and symptoms of diarrhea, diarrhea causes, preventive measures, and critical signs of diarrhea, as well as critical signs of dehydration. The third part focused on the manner of management practice of diarrhea at home. The fourth part was the attitude domain questions that were composed of 11 questions assessed by a 5-point Likert Scale (strongly agree, agree, neutral, disagree, and strongly disagree). A panel of three members (two professors and a researcher) from the College of Pharmacy, King Saud University, who were experts in preparing the study tool were reviewed the questionnaire. The survey was translated into Arabic by an independent translator, and reviewed again for the appropriateness of language before testing its validity. The questionnaire was then validated through randomly selected 10 respondents in a pilot study carried out at the College of Pharmacy, King Saud University. The respondents recruited in the pilot study were mothers and did not include them in the final results or had no contact with the subjects of the study. The reliability test was determined using Cronbach’s alpha of the questionnaire for knowledge (0.73), attitudes, and (0.71) suggested that questionnaires can be used to carry out the study.

### 2.3. Participants Recruitment Procedure

The validated Arabic questionnaires were used for data collection. Social media platforms were chosen as the potential medium for data collection using the snowball technique. For the purpose of data collection, a team of three female students were appointed from the female campus of the College of Pharmacy, King Saud University, and given clarity about inclusion and exclusion criteria and the procedure of data collection through social media platforms. The data collators were strictly investigated by two senior academicians of the pharmacy college. The data collectors had to ensure that the questionnaires reached all the regions of Saudi Arabia. The questionnaires started with a pre-condition that stated that “Saudi females with at least one child are only eligible to fill in the questionnaire; females without children are not allowed to complete the questionnaires. During the data collection period, we received a total of 1140 responses. According to the timeline of the data collection period, we stopped receiving the responses once the timeline end.

### 2.4. Sample Size Estimation

According to previous reports, the prevalence of diarrhea among pediatrics was 56.3% [16]. The sample size for the given study was calculated by using the following Equation (1):*n* = z^2^ × p × q/d^2^(1)
where *n* is the minimum sample size, z is the constant (1.96), p is the prevalence of diarrhea (among pediatrics, it was 0.563%), q is (1 − p), Z is the standard normal deviation of 1.96, corresponding to the 95% confidence interval, and d is the desired degree of accuracy.
*N* = (1.96)^2^ × 0.563 × (1 − 0.563)/(0.05)^2^*n* = 378

### 2.5. Data Analysis

Statistical Package for Social Sciences version IBM SPSS Statistics 26 (IBM Inc., Chicago, IL, USA) software was applied to analyze the data. Descriptive statistics, including percentages and frequencies, were used to present the results.

## 3. Results

### 3.1. Sample Characteristics

A total of 1140 respondents filled the questionnaire. About 24% of respondents were aged from 36 to 40 years, and only 5.4% of respondents received no formal education. More than half of mothers were housewives (52.1%). Slightly more than one-third of children were aged above two years. The demographic of mothers and their children are summarized in Table 1.

### 3.2. Mothers’ Knowledge about Childhood Diarrhea

In this study, the mother rated the critical signs of childhood diarrhea as follows: Blood in the stool (49%), followed by thirst and dry mouth (32.1%), and loss of stretchiness of the skin (24.3%). Nearly half (*n* = 569; 49.9%) of mothers believed that diarrhea is caused by eating dirty food, while (*n* = 516) 45.3% of them thought that eating expired food; also, (*n* = 470) 41.2% of them believed that eating with dirty hand cause the transmission of diarrhea. Furthermore, a detailed description of mothers’ knowledge about childhood diarrhea was given in Table 2.

In this study, the most reported practice among mothers towards management of their child’s diarrhea included seeing the physicians (68.9%), giving homemade fluids (52.6%), using ORS (23.0), and visiting pharmacists (20.4) (Table 2).

### 3.3. Mothers’ Knowledge about the Use of ORS

In this study, about 62% of mothers know the ORS, but only 53.5% of them recognized that ORS could prevent the child from getting dehydrated. The resources of information for utilizing ORS in childhood diarrhea were gained mainly from medical prescription (50.3%), followed by family (15.7%), and then by consultation of pharmacists (15.1%) as presented in Table 3.

### 3.4. Mothers’ Attitude about Childhood Diarrhea Mortality and Treatment

In this study, around 33.5% of mothers thought diarrhea can attack bottle-feed children. The majority of mothers (65.9%) reported that teething is the main cause of diarrhea. The disagreement that “liquid food aggravates diarrhea” were reported by 22.5% of the mothers. About 41% mothers thought that diarrhea is a problem in the Saudi community, and 60.9% of mothers stated that handwashing prevents diarrhea. More details on mothers’ attitude on childhood diarrhea are shown in Table 4.

## 4. Discussion

This study assessed the mothers’ knowledge on the critical signs of diarrhea. About 49% of the mothers stated that the passage of three or more loose stools with blood during the day is an obvious critical sign of diarrhea that require hospital or physician visit, and almost 23% reported no knowledge of any critical sign of diarrhea. These results showed low knowledge among Saudi mothers about the critical signs of diarrhea, but their knowledge of the critical signs of diarrhea was higher compared to other mothers in different communities. A study carried out in Ethiopia found that 39.5% of mothers stated that passage of three or more loose stools with blood during the day is the mark of severe diarrhea [17]. A similar study was done in the rural setting of Kenya determined that majority of mothers (76.4%) did not identify the critical signs of childhood diarrhea [18]. A study from Nepal reported that 20.8% of mothers identified red-colored diarrhea as “the most dangerous diarrhea [2]. The critical signs of dehydration have the similar importance to bloody diarrhea, and the participating mothers in our study were unable to identify the most common signs of dehydration. Only 32.1%, 11.6%, and 24.3% were able to recognize thirst/dry mouth, tearless eyes, and loss of strictness of skin, respectively, as the critical signs of dehydration secondary to diarrhea. The poor knowledge and unrecognized signs of dehydration among mothers are noticed globally [2,19,20,21]. Adequate knowledge about the critical signs of childhood diarrhea is essential as the early referral of a child with severe diarrhea is fundamental for appropriate treatment [17].

Poor sanitation of food and water may lead to diarrhea, with approximately 3000 deaths and 135,000 hospitalizations annually secondary to the food-borne transmission of diarrhea causative pathogens in the United States [22]. In regards to the causes and transmission of diarrhea, nearly 50% of participating mothers believed that eating polluted food is the most common cause of childhood diarrhea, and only 31% of mothers identified drinking unclean water as the reason for childhood diarrhea. In consistent with our findings, many studies reported a low level of mothers towards causes and transmission of diarrhea. [2,17,23]. A study from Nigeria reported that the most common causes for childhood diarrhea were contaminated food (24.1%) and unclean water (11.3%) [8]. Another study from Iran showed that only 24.66% of mothers knew that unclean water can cause diarrhea [24]. A study was carried out in Malawi reported that 55% of mothers stated that unhealthy water is the main cause of diarrhea [25]. The change in knowledge of childhood diarrhea could be due to variance in mothers’ education levels.

The misconception between teething and diarrhea seems disseminated widely. The results of the present study showed that 66% of mothers stated teething is a reason for their child’s diarrhea. These results come agreed with other studies from different countries [8,24]. In addition to the wrong belief of association between diarrhea and teething, mothers tend to consider diarrhea secondary to teething as “non-serious diarrhea”, and they may deal with it loosely even if it’s accompanied by critical signs like dehydration [26]. The efforts should be applied to educate the mothers about the critical signs of diarrhea in children, and to disassociate the belief link between diarrhea and teething.

The mothers’ sufficient knowledge on reasons, prevention, and management of diarrhea utilizing proper therapies is the key for home management of childhood diarrhea [14,27]. According to the Integrated Management of Childhood Illness (IMCI) guidelines, the use of ORS is the principle therapy of diarrhea [28]. However, the use of ORS seemed not highly encouraging among mothers. Although 62% of our participating mothers knew about the ORS, only 23.5% of them used it for their children. Similar findings were reported. A study from Nigeria reported that most mothers (63%) were aware of ORS, but 27% of them used it for their children. In another study did in Pakistan mentioned that 58% of mothers used ORS to treat their childhood diarrhea disease [12]. Moreover, in our study, it was found the main resources of mothers regarding ORS usage were medical prescriptions (50.3%), family/relatives/friends (15.7%), and pharmacists (15.1%). In a similar study, the two main resources of ORS information were families/friends (76%) and pediatricians (58%) [29]. Despite 53.3% of our participants reporting that “ORS prevents a child from getting dehydrated”, few mothers are using ORS (23.5%). The reason for a few mothers used ORS could be due to their level of education and awareness of the mothers on diarrhea management, due to inadequate public information on this issue.

This study was conducted online to diminish geographical dependence, and because of the high rate of literacy and social media usage among Saudis [30,31], as well as this study’s aim to assess the knowledge, attitude, and practice of mothers towards childhood diarrhea from a community perspective. The study was performed with significance that mothers’ knowledge of childhood diarrhea prevention and management would decrease the unnecessary hospital or clinic visits; however, most of the participating mothers (68.9%) seek treatment from a medical doctor for their children with diarrhea. This practice of mother was observed in a previous study [17]. In contrast, our subjects showed good preventive measures. The majority of mothers (70.9%) believed that handwashing, which is one essential measure to decrease the prevalence of diarrhea [32], prevents childhood diarrhea.

## 5. Conclusions

Approximately 40.3% of participating mothers believed that childhood diarrhea is a major problem in the Saudi community. These beliefs are supported by reports mentioned the prevalence of diarrhea is high in some areas of Saudi Arabia [1]. However, insufficient knowledge of childhood diarrhea and its management were observed. Adequate knowledge about the critical signs, causes, transmission, prevention, and management of childhood diarrhea should be applied in simple language to communicate the health-related information clearly. Moreover, there is a lack of knowledge on the role of ORS and its use. The general understanding of ORS is not adequate and needs reliable efforts to highlight its importance in resolving dehydration of childhood diarrhea.

## Figures and Tables

**Table 1 ijerph-18-03982-t001:** Demographic data of mothers and their children.

Variables	Number	Percentage
Age of the Mothers (years) *		
18–20	130	11.4
21–25	130	11.4
26–30	166	14.6
31–35	192	16.8
36–40	271	23.8
more than 41 years	248	21.8
Education level		
Illiterate	62	5.4
Primary school/secondary school	160	14
High school	244	21.4
University	479	42
Postgraduate	195	17.1
Job		
Employer	427	37.5
Health staff	68	6
Housewife	594	52.1
Students	39	3.4
Other’s	12	1
Insurance		
None	329	28.9
Governmental	482	42.3
Private	329	28.9
Age of the child (years)		
Less than on year	217	19
1–2 years	226	19.8
Above 2 years	697	61.1
Gender of a child		
Male	456	40
Female	675	59.2

* Missing Data.

**Table 2 ijerph-18-03982-t002:** Mothers’ knowledge about diarrhea and its management.

Variables	*N*	%
Critical signs and symptoms of diarrhea *		
Passage of >3 loose stools with blood in 24 h	553	48.5
Thirst and dry mouth	366	32.1
Tearless eyes	132	11.6
Loss of stretchiness of the skin	277	24.3
I do not know	261	22.9
Cause/mode of transmission of diarrhea *		
Drinking bad/dirty water	361	31.7
Eating dirty food	569	49.9
Eating with dirty hands	470	41.2
Eating expired food	516	45.3
Insects feeding on dirty tools	231	20.3
Bottle of breastfeeding	235	20.6
Unpasteurized milk	226	19.8
Management of child’s diarrhea		
Visit the physician	786	68.9
Visit the pharmacist	233	20.4
Use ORS	262	23
Give homemade fluids	600	52.6
Feed the child with breastfeeding	100	8.8
Stop feeding	53	4.6
Nothing	41	3.6

* Multiple-answers question.

**Table 3 ijerph-18-03982-t003:** Mothers’ knowledge about the use of ORS.

Characteristic	Number	Percentage
Do you know the ORS?		
Yes	706	61.9
No	434	38.1
Role of ORS solution in diarrhea		
Prevents child from getting dehydrated	610	53.5
Either increases or decreases diarrhea	183	16.1
No role in diarrhea treatment	77	6.8
Sources information of use ORS		
Medical prescription	573	50.3
Consultation of pharmacists	172	15.1
Family	179	15.7
Internet	112	9.8
TV	31	2.7

**Table 4 ijerph-18-03982-t004:** Mothers’ attitude about childhood diarrhea.

Questionnaires	Strongly Agree *n* (%)	Agree *n* (%)	Neutral *n* (%)	Disagree *n* (%)	Strongly Disagree *n* (%)
Diarrhea attacks mostly bottle-feed children	146 (12.8)	236 (20.7)	522 (45.8)	180 (15.8)	56 (4.9)
Diarrhea is a disease of the poor children	36 (3.2)	49 (4.3)	223 (19.6)	431 (37.8)	401 (35.2)
Diarrhea is a problem in the community	124 (10.9)	347 (30.4)	411 (36.1)	190 (16.7)	68 (6.0)
Teething causes diarrhea	282 (24.7)	470 (41.2)	239 (21.0)	105 (9.2)	44 (3.9)
Diarrhea is a curable disease	559 (49.0)	449 (39.4)	81 (7.1)	30 (2.6)	21 (1.8)
Liquid food aggravates diarrhea	111 (9.7)	243 (21.3)	529 (46.4)	209 (18.3)	48 (4.2)
Oral rehydration salts solution cures diarrhea	141 (12.4)	368 (32.3)	485 (42.5)	122 (10.7)	24 (2.1)
Human feces are a source of diarrhea	120 (10.5)	196 (17.2)	648 (56.8)	133 (11.7)	43 (3.8)
Handwashing prevents diarrhea	248 (21.8)	446 (39.1)	264 (23.2)	122 (10.7)	60 (5.3)

## Data Availability

Data will be available upon request from the corresponding author of the study.

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
