# Peer review of "Assessment of Saudi Mother’s Knowledge and Attitudes towards Childhood Diarrhea and Its Management"

_ijerph, 2021, doi:10.3390/ijerph18083982_

Round 1

Reviewer 1 Report

This is an interesting study of the knowledge and practices of mothers with children with diarrhea conducted in Saudi Arabia. The study provides some interesting information, however there are several areas that would benefit from further clarification or revision.

In the abstract, the purpose is noted as demonstrating the best care of children with diarrhea by investigating knowledge deficiencies and improper practices. If the purpose is to describe the best care, this would be phrased differently. It appears that the purpose is to determine the knowledge and practices rather than demonstrate best practices.

The introduction is adequate. The purpose statement at the end of the introduction has the same issue as it does in the abstract.

The design notes the use of a cross-sectional survey of mothers in Saudi Arabia. It says is it an on-line survey in the abstract. More information is needed about the recruitment and how the cross-sectional sample was determined. Recruitment methods are not described nor is it clear how many surveys were sent out to determine the return rate.

The design of the surveys is well described. The testing of the survey appears to include a sample of 10 which is appropriate for feedback and face validation. The Cronbach’s alpha analysis is not clear as the sample size of 10 inadequate for analysis by Cronbach’s alpha.

The data analysis notes Chi-square analysis but the comparison did not appear to be included in the manuscript.

Results are missing the return rate of the surveys. It is noted that 5.4% of the respondents were illiterate and is it not clear how they completed an on-line written survey. The number of children may be a factor in the gender of the children as they could have male, female or both in the family.  The presentation of the data in the tables is clear, however the text appears to point out data that demonstrates a lack of knowledge or less than desirable practices but fails to note that over 20% of the mothers are more than 41 years old and children may be nearly grown and knowledge and practices may have changed. An analysis of the younger mothers may present a better representation of the current knowledge and practices. This information should be noted in the results or the discussion.

The discussion compares results to surveys from developing countries as well as the United States. These seem to be somewhat extreme comparisons and it is difficult to interpret since the sample is not clearly defined.

Limitations are missing from the manuscript.

There is not a review by an ethics committee noted and a lack of description of recruitment methods. These need to be included in the manuscript.

There are a number of grammatical, punctuation and spelling errors that need to be corrected.

Author Response

This is an interesting study of the knowledge and practices of mothers with children with diarrhea conducted in Saudi Arabia. The study provides some interesting information, however there are several areas that would benefit from further clarification or revision. In the abstract, the purpose is noted as demonstrating the best care of children with diarrhea by investigating knowledge deficiencies and improper practices. If the purpose is to describe the best care, this would be phrased differently. It appears that the purpose is to determine the knowledge and practices rather than demonstrate best practices.

My apologizes I have edited it was by mistake , due to multiple studies going with us

The introduction is adequate. The purpose statement at the end of the introduction has the same issue as it does in the abstract.

My apologies I have corrected it

The design notes the use of a cross-sectional survey of mothers in Saudi Arabia. It says is it an on-line survey in the abstract. More information is needed about the recruitment and how the cross-sectional sample was determined. Recruitment methods are not described nor is it clear how many surveys were sent out to determine the return rate.

The validated Arabic questionnaires were used for data collection. Social media plat-forms were chosen as the potential medium for data collection using snowball technique. For the purpose of data collection, a team of three female students were appointed from the female campus college of pharmacy, king Saud university, and given the clarity about inclusion and exclusion criteria and the procedure of data collection through social media platforms. The data collators were strictly investigated by two senior academicians of the pharmacy college. The data collectors had to ensure that the questionnaires reached all the regions of Saudi Arabia. The questionnaires started with a pre-condition that stated that “Saudi females with at least one child are only eligible to fill in the questionnaire; females without children are not allowed to complete the questionnaires. During the data collection period, we received a total of 1140 responses. According to the timeline of the data collection period we stop to receive the responses once the timeline end.

The design of the surveys is well described. The testing of the survey appears to include a sample of 10 which is appropriate for feedback and face validation. The Cronbach’s alpha analysis is not clear as the sample size of 10 inadequate for analysis by Cronbach’s alpha.

The data analysis notes Chi-square analysis but the comparison did not appear to be included in the manuscript.

Statistical Package for Social Sciences version 25 (SPSS) software was applied to analyze the data. Descriptive statistics including percentages and frequencies were used to present the results

Results are missing the return rate of the surveys. It is noted that 5.4% of the respondents were illiterate and is it not clear how they completed an on-line written survey. The number of children may be a factor in the gender of the children as they could have male, female or both in the family.  The presentation of the data in the tables is clear, however the text appears to point out data that demonstrates a lack of knowledge or less than desirable practices but fails to note that over 20% of the mothers are more than 41 years old and children may be nearly grown and knowledge and practices may have changed. An analysis of the younger mothers may present a better representation of the current knowledge and practices. This information should be noted in the results or the discussion.

We have randomly distributed the survey to Saudi mothers, included in the questionnaires, mother who are having at least one child eligible fill the survey, there was an option, proceed or end the survey. we distributed approximately 2000 questionnaires online, from those 1140 were answered completely and return the survey.

5.4% of the respondents were illiterate: yes, they might take the help of their family members to fill the survey,

The discussion compares results to surveys from developing countries as well as the United States. These seem to be somewhat extreme comparisons and it is difficult to interpret since the sample is not clearly defined.

We have corrected it

Limitations are missing from the manuscript.

We have included it

There is not a review by an ethics committee noted and a lack of description of recruitment methods. These need to be included in the manuscript.

We have included it

There are a number of grammatical, punctuation and spelling errors that need to be corrected.

We have included it

Reviewer 2 Report

I think you should try stratifying by education level to determine whether misconceptions predominate in lower educational levels of mothers compared to higher ones.  If you find that misconceptions are present regardless of educational leve, you speak to a need for greater public education on the importance of ORS in treating diarrhea. 

Author Response

I think you should try stratifying by education level to determine whether misconceptions predominate in lower educational levels of mothers compared to higher ones.  If you find that misconceptions are present regardless of educational level, you speak to a need for greater public education on the importance of ORS in treating diarrhea. 

My apologies, our study objective was to evaluate the knowledge and attitudes of Saudi mother about diarrhea and its management

Reviewer 3 Report

I have read this paper with great interest, and do have 3 type of comments

first and minor, the editing of table 4, and the writing/grammar in the paper should be further considered. 

second, We need more background information on the setting in this country: languages ? socio-economics of those who contribute to the questionnaire compared to the full population ? and how were mothers targeted (at random, specific age categories of their children ?)Was the number of children in the household also registered ?

finally, and most relevant, the validity of the questionnaire should be further considered or explored ? if I understood this correct, the construct validity was assessed only by pharmacists ? somewhat bizar, why were no nurses or MDs involved; like eg 3 bloody stools in the list, but fever ? sleepy or less playful ?

Author Response

I have read this paper with great interest, and do have 3 type of comments

first and minor, the editing of table 4, and the writing/grammar in the paper should be further considered. 

my apologies i have corrected it

second, we need more background information on the setting in this country: languages? socio-economics of those who contribute to the questionnaire compared to the full population? and how were mothers targeted (at random, specific age categories of their children?) Was the number of children in the household also registered?

my apologies i have corrected it in method section

As due to the nature of Arabic in Saudi Arabia, we used English questionnaires and translated in Arabic using forward and backward translation procedure. And we included mother bearing at least one child aged below 5 years  

finally, and most relevant, the validity of the questionnaire should be further considered or explored? if I understood this correct, the construct validity was assessed only by pharmacists? somewhat bizar, why were no nurses or MDs involved; like e,g 3 bloody stools in the list, but fever? sleepy or less playful?

My apologies, we have validated this questionnaires using random sample of 10 Saudi mothers bearing at least one child and we have research team, for the data collection, team of 3 Saudi female students from female campus, they targeted the sample. study questionnaires were evaluated by senior professor from college of pharmacy for the face and content validity. This study is qualitative study collection aptients knwoldge and attitudes, so we didn’t include nurses or MD because senior professor who had PhD in clinical pharmacy and giving the consultation to the patients

Round 2

Reviewer 1 Report

Concerns have been addressed. 

Author Response

Concerns have been addressed. 

Dear reviewer and editorial team my sincere apologies for the errors if any 

Thank you very much for your response and for sending back the comment 

Reviewer 2 Report

I suggest that you use wording emphasizing that you are establishing the gaps in the knowledge base of your sample. You bring up that mothers really are not aware of unclean water as a source of transmission of diarrhea. What are the ramifications? Why should we work on addressing the knowledge and attitude gaps? This is the part of your paper that is novel and original, and you need to emphasize this more in your discussion. Otherwise, the readers' interest will remain average. You make comparisons with other populations, but you need to tie them together more so that your readers see that the lessons learned from the Saudi mothers are generalizable to other cohorts.  

There are three parts to your educational objective:

Saudi mother's knowledge of the causes of diarrhea

attitudes towards the potential mortality of diarrhea

attitudes towards the treatment of diarrhea

You need to state all 3 clearly and separately. you mix prepositions by bundling them into one sentence. 

Author Response

Reviewer 2

I suggest that you use wording emphasizing that you are establishing the gaps in the knowledge base of your sample. You bring up that mothers really are not aware of unclean water as a source of transmission of diarrhea. What are the ramifications? Why should we work on addressing the knowledge and attitude gaps? This is the part of your paper that is novel and original, and you need to emphasize this more in your discussion. Otherwise, the readers' interest will remain average. You make comparisons with other populations, but you need to tie them together more so that your readers see that the lessons learned from the Saudi mothers are generalizable to other cohorts. 

Dear editor and team, thank you very much for the comments and I have done separating as per your suggestions.  In this current study about one third 31.7% of the participants agreed Drinking bad/dirty water will cause the diarrhea. This might be due to fact that in Saudi Arabia, ministry of health and government follow very strict conditions towards foods and supplies, and efforts made to unavailability or visibility of dirty water anywhere in the city. Although in ksa, all the water available in the form of bottled packed filtered water, there are very rare circumstances that bad or expired water may found in the ksa

Why should we work on addressing the knowledge and attitude gaps?

There was numerous study in different discipline, revealed that adequate knowledge and attitudes towards diseases, reflect the disease morbidity and mortality, persons with enough knowledge and attitude will have more secure from the disease. Measuring the gap will add the literature furthermore baseline for the research. Therefore, health care professionals and health system they will get an idea and perception of the patient towards the disease. There are three parts to your educational objective:

Saudi mother's knowledge of the causes of diarrhea

attitudes towards the potential mortality of diarrhea

attitudes towards the treatment of diarrhea

Yes, separated in results

You need to state all 3 clearly and separately. you mix prepositions by bundling them into one sentence. 

Yes, separated in results. My apologies I have rewritten with separate headings

Reviewer 3 Report

thank you for the revised version of your paper as the methods are in my opinion better described. I do have however still one aspect that has not yet been handled sufficiently well: how do the respondents 'compare' to the 'average' mothers in the country, when related to age, number of children, education etc, how 'confident' are you that the group of respondents are representing the targeted population ? are there biases ? 

Author Response

Reviewer 3

thank you for the revised version of your paper as the methods are in my opinion better described. I do have however still one aspect that has not yet been handled sufficiently well: how do the respondents 'compare' to the 'average' mothers in the country, when related to age, number of children, education etc., how 'confident' are you that the group of respondents are representing the targeted population? are there biases? 

Dear editor in chief and reviewer team, thank you very much for the review comment, and my sincere apologies for the errors if any. we have mainly targeted Saudi mothers with at least one child, for this purpose we appointed a team of female researchers, for collecting the data. In Saudi Arabia, most marriages occur at a younger age and the childbirth rate is high. We have mentioned in the first question, that this study belongs to Saudi women bearing at least one child, are eligible to fill and participate in the survey, by this way we collected data. We are full of confidence that this respondent was targeted for the study in Saudi Arabia  
